# Sweeteners’ Influence on In Vitro α-Glucosidase Inhibitory Activity, Cytotoxicity, Stability and In Vivo Bioavailability of the Anthocyanins from Lingonberry Jams

**DOI:** 10.3390/foods12132569

**Published:** 2023-06-30

**Authors:** Teodora Scrob, Gabriela Adriana Filip, Ioana Baldea, Sânziana Maria Varodi, Claudia Cimpoiu

**Affiliations:** 1Faculty of Chemistry and Chemical Engineering, Babes-Bolyai University, 11 Arany Janos, 400028 Cluj-Napoca, Romania; teodora_scrob@yahoo.com (T.S.); sanzianavarodi@yahoo.com (S.M.V.); 2Research Center for Advanced Chemical Analysis, Instrumentation and Chemometrics, 11 Arany Janos, 400028 Cluj-Napoca, Romania; 3Department of Physiology, “Iuliu Hatieganu” University of Medicine and Pharmacy, 1-3 Clinicilor Street, 400006 Cluj-Napoca, Romania; adrianafilip33@yahoo.com (G.A.F.); baldeaioana@gmail.com (I.B.)

**Keywords:** anthocyanins, lingonberry jams, sweeteners, bioavailability, α-glucosidase, cytotoxicity

## Abstract

Several lines of evidence demonstrate the multiple health-promoting properties of anthocyanins, but little is known regarding the bioavailability of these phytochemicals. Therefore, the stability during storage and bioavailability of anthocyanins from lingonberries jams were determined by HPLC, together with the impact of used sweeteners on their adsorption. Further, the in vitro α-glucosidase inhibition using spectrophotometric methods and cytotoxicity determined on normal and colon cancer cells were communicated. The content of anthocyanins was significantly decreased during storage in coconut sugar-based jam, but was best preserved in jam with fructose and stevia. Fructose and stevia-based jams showed the highest inhibition activity upon α-glucosidase. Lingonberry jams showed no cytotoxic effects on normal cells, but at low concentration reduced the tumor cells viability. Anthocyanins were still detectable in rats’ blood streams after 24 h, showing a prolonged bioavailability in rats. This study brings important results that will enable the development of functional food products.

## 1. Introduction

Anthocyanins are a major class of phytochemicals, predominantly present in red berries, grapes or red wine, that gained attention due to their multiple health-promoting properties [1,2]. A lower incidence of developing cardiovascular diseases is correlated with the consumption of anthocyanins-based foods [3]. The chemical structure of anthocyanins contains one unit of anthocyanidins (pelargonidin, cyanidin, peonidin, delphinidin, petunidin and malvidin) and one sugar moiety, such as glucose, rhamnose, galactose or arabinose [2,4]. Anthocyanidins are responsible for the color of fruits and vegetables. For example, delphinidin confers the blue color, whereas red and purple colors are related to cyanidin and pelargonidin contents [5]. As a consequence, due to their food-coloring properties, anthocyanins represent also important substituents for synthetic food dyes and colorants. Unfortunately, these molecules are not very stable, being highly sensitive to light and temperature [3]. For these reasons, the study of anthocyanins under different conditions became a challenge not only for the food industry, but also for other interested decision-makers due to their high instability during storage.

A central aspect of this study is that not all the anthocyanins are absorbed after they are ingested through diet. It was reported that the absorption of these pigments is low [3], and their health-related functions may be reduced. Due to their sensitivity at varying pH, anthocyanins structures are exposed to various changes during digestion. It was shown that anthocyanins are more stable under acidic pH [6], but under alkaline pH, anthocyanins undergo limited absorption [3]. Thus, in order to evaluate their biological activity within target tissues, the study of their bioavailability is necessary. Bioavailability studies of phytochemicals such as anthocyanins can be conducted by both in vitro and in vivo digestion models. The use of in vivo studies is largely used nowadays, covering the gap between in vitro processes and human organisms. The most used animals for in vivo studies are rodents (rats and mice), due to the fact that the digestion process of rodents is very close to that of human, including absorption, metabolism and excretion. In addition, the use of rodents in in vivo studies is easy to handle, low cost and available [7].

Besides their multiple benefits, anthocyanins are also involved in regulating carbohydrate digestion by inhibiting the activity of digestive enzymes, such asα-glucosidase [8]. In this way, the control of postprandial plasma glucose levels is very important, both for early treatment of diabetes mellitus and for the reduction of chronic vascular complications [9]. The enzyme α-glucosidase from the small intestine is involved in hydrolyzing small reducing sugars molecules to a single α-glucose molecule [8]. The α-glucosidase inhibits the carbohydrate absorption in the small intestine and influences the insulin levels. Although acarbose is commercially available, several α-glucosidase inhibitors have been isolated from natural sources [9], such as lingonberries, rich in anthocyanins.

A number of studies have examined the chemopreventive and antitumoral effects of anthocyanins. The chemopreventive mechanisms include scavenging-free radicals, ability to reduce the cell proliferation or up-regulating/triggering the apoptosis [10]. Since the use of natural products in preventing and treating several affections is more and more popular, finding chemopreventive agents from natural sources may play an important role nowadays. It was reported that anthocyanins and their aglycones, such as cyanidin, delphinidin, malvidin, pelargonidin and peonidin, exhibit antiproliferative and proapoptotic properties in Caco-2 (colon cancer) cells [10]. However, further studies with different doses and on various experimental models are necessary for the assessment of benefits in oncology.

It was reported that the food matrix affects the bioavailability of anthocyanins, and, consequently, some biological effects, such as antidiabetic activity or cytotoxicity, can be affected. Various phytochemicals can interact with other components of food, such as proteins, polysaccharides or sugars, and these interactions may limit the efficiency of absorption. Ortega et al. [11] reported that the fat content of cocoa may enhance the digestibility of procyanidins. Therefore, studying the effect of the food matrix on the digestibility and bioavailability of anthocyanins is an important step in developing new food formulas and functional beverages. In this line, the main objective of this research was to investigate the bioavailability of anthocyanins from lingonberry jams formulated with different sweeteners in an in vivo model. In addition, their potential in inhibiting α-glucosidase activity, as well as the cytotoxicity induced by jams on human dermal fibroblasts (BJ) and colon cancer cells (Caco-2), was also investigated. This study focuses also on evaluation of the changes in the anthocyanin profile in the presence of different sweeteners during 180 days of storage, as a future perspective for functional foods development.

## 2. Results and Discussion

### 2.1. Effect of Sweeteners on the Stability of Anthocyanins from Lingonberry Jams during Storage

Among phytochemicals present in lingonberries, anthocyanins play an important role. Even though anthocyanins possess antioxidant properties, they have the disadvantage of being sensitive to exposure to light, high temperatures or even acids [3], and evaluating the factors influencing their stability is important. Three main anthocyanins were identified by HPLC in all jams, namely, cyanidin-3-galactoside (Peak 1), cyanidin-3-glucoside (Peak 2) and cyanidin-3-arabinoside (Peak 3) (Figure 1). This finding is in agreement with the results obtained by other researchers [12,13,14].

Among the three anthocyanins identified in lingonberry jams, cyanidin-3-galactoside was found to be the most abundant anthocyanin (84.61%), followed by cyanidin-3-arabinoside (9.76%) and cyanidin-3-glucoside (5.63%). The content of each anthocyanin is highly influenced by different storage conditions (Table 1 and Appendix A).

Following 180 days, anthocyanin preservation in jams stored at 4 and 25 °C under light ranges from 51.6–68.6% to 0.10–0.51%, respectively. This finding suggests that storage at lower temperatures and in absence of light results in better anthocyanins preservation, the results being in agreement with previously reported data for black carrot jams [15]. During storage at 25 °C under dark, individual anthocyanins present a higher stability than during storage under light at the same temperature. Anthocyanins are very sensitive molecules in the presence of light due to their degradation into phenolic acids and aldehydes [15].

Each anthocyanin is influenced also by the type of sweetener used in jam formulations (Appendix A), regardless of storage conditions. Anthocyanins from jam containing fructose are preserved better than in the case of jams prepared with the other sweeteners. In the case of fructose-based jam, cyanidin-3-galactoside decreased from 95.1% after 15 days of storage to 67.6% after 180 days of storage under refrigeration (Table 1, Figure 2a). As regarding cyanidin-3-glucoside and cyanidin-3-arabinoside, their decrease ranged between 62.9% and 68.6%, respectively. However, in the presence of light, cyanidin-3-galactoside and cyanidin-3-arabinoside loss reached up to 99.9% at the end of storage, whereas cyanidin-3-glucoside was not detected anymore (Figure 2b), anthocyanins being more susceptible to degradation under light [3]. For fructose-based jam stored under dark at 25 °C, the loss was higher than 98% in the case of all 3anthocyanins (Figure 2c), indicating that regardless of the sweetener, a high temperature may negatively impact the stability of anthocyanins.

Under refrigeration, erythritol also seemed to protect cyanidin-3-galactoside in a higher proportion than other sweeteners, its decrease reaching up to 64.5% after 180 days, followed by cyanidin-3-arabinoside and cyanidin-3-glucoside. However, erythritol-based jams determine a decrease of 100% of all 3anthocyanins in the case of samples stored under light and dark at 25 °C during 180 days. This fact indicates that some sweeteners may exert a protective effect upon anthocyanins stability only at lower temperatures, whereas under heat and light conditions, an opposite effect might be observed. The same pattern can be also observed in the case of stevia- and saccharine-based jams (Table 1).

In the presence of saccharine, cyanidin-3-galactoside decreased up to 66.9%, whereas in the presence of both white and brown sugar, its decrease reached up to 61.3%. This fact indicates that sugar does not protect, as do other sweeteners, the anthocyanins in lingonberry jams. Interestingly, in a study reported by Kamiloglu et al. [15] on black carrot jams, samples prepared with sugar retained a higher amount of individual anthocyanins compared to jams and marmalades with sweetener, most probably due to the reduced water activity. This finding suggests the importance of the food matrix in the stability of anthocyanins during storage.

Coconut sugar is the sweetener that determines the highest decrease of cyanidin-3-galactoside under refrigeration: 20.5% (Table 1). Interestingly, coconut sugar seemed to have a higher protection upon the other two anthocyanins. Nevertheless, compared to jams formulated with the other sweeteners, samples prepared with stevia retained the highest amount of cyanidin-3-galactoside (77.5%) after 180 days of storage under refrigeration. However, stevia did not protect the same the other two anthocyanins, their decrease ranging between 51.6 and 55.8%, respectively. These results indicate that sweeteners may protect better a specific type of anthocyanin.

The results of this study show that cyanidin-3-glucoside is the anthocyanin with the highest instability during storage, regardless of the sweetener used or storage conditions, followed by cyaninidin-3-galactoside and cyaninidin-3-arabinoside. Therefore, we can conclude the sugar moiety of anthocyanin influencing these pigments’ degradation. The main reason for anthocyanin degradation is hydrolysis of the glycoside linkage and the presence of the flavylium salts in acidic conditions [15]. This suggestion correlates well with results reported in our previous study [16], where jam prepared with coconut sugar presented the highest level of total acidity after storage. In this study, coconut sugar jam presented the highest loss of anthocyanins, indicating a higher stability of flavylium salts in this sample.

From the obtained results, it is evident that individual anthocyanins are mostly affected by light and the higher temperature of storage. On the other hand, the type of sweetener used in jam formulation also seemed to affect positively or negatively anthocyanins behavior during storage. The sensitivity of anthocyanins was confirmed in this research by decreases after only 15 days of storage (Table 1).

### 2.2. Inhibitory Activity of Lingonberry Jams against α-Glucosidase

The ability of lingonberry jams extracts to inhibit in vitrothe α-glucosidase has been evaluated using an enzymatic method [9]. In humans, dietary carbohydrates are hydrolyzed by the intestinal α-glucosidase enzyme, which is responsible for the breakdown of complex sugars into monosaccharides, a form suitable for absorption. The inhibition of this enzyme will delay release of glucose into the blood and can be useful for diabetes treatment [8,9]. Acarbose inhibits the enzyme α-glucosidase [9] and diminishes the postprandial blood glucose. Thus, acarbose is used to treat diabetic patients. For this reason, acarbose is usually used as a positive control. A higher degree of inhibition of α-glucosidase is correlated with a higher antidiabetic activity, because the activity of the enzyme is interrupted and glucose is no longer released into the blood. The inhibition degrees of lingonberry jams on α-glucosidase were determined, and the results are comparatively presented in Figure 3 and in Appendix A. As is expected, the lowest inhibition degree of the enzyme was determined in the case of jams with sweeteners such as sugar and brown sugar. On the contrary, the highest inhibition degree values were found in the case of fructose- and stevia-based jams (up to 43%). Moreover, for a better understanding of sweeteners’ effect, the analysis was also performed on the jam prepared without any sweetener, for which the highest inhibition degree (57%) was obtained. These results suggest that all the sweeteners may statistically significantly influence the inhibitory activity of lingonberry jams against α-glucosidase (*p* < 0.05), decreasing its values. Further, the inhibitory activity of Jam 1 does not differ statistically significantly from those of Jams 4 and 7, that of Jam 3 from those of Jams 5 and 6 and that of Jam 6 from those of Jams 2 and 5, in all cases *p* values being higher than 0.05.

The equivalent concentration of acarbose (µg/mL) was also determined based on a calibration curve (Appendix A). The highest value of the equivalent acarbose concentration is observed in the case of unsweetened jam (132.1 µg/mL), followed by Jam 2 (80.3 µg/mL), Jam 6 (71.3 µg/mL), Jam 5 (50.8 µg/mL), Jam 3 (50.2 µg/mL) and Jam 7 (10.1 µg/mL). In the case of white and brown sugar jams, the equivalent acarbose concentration is 0.0 µg/mL. This fact indicates that jams prepared with white and brown sugar do not have an equivalent concentration of acarbose, and these sweeteners are not recommended for diabetic consumers. From the results obtained in this study, we can conclude that 1 mL of fructose-based lingonberry jam extract is equivalent to 80.2 µg acarbose, whereas 1 mL of stevia-based lingonberry jam extract is equivalent to 71.3 µg of acarbose.

Generally, the normal daily dose of acarbose recommended for a diabetic person is between 150 and 300 mg acarbose [17]. Thus, consuming 113.5 g unsweetened lingonberry jam would be equivalent to the recommended daily dose of acarbose. In the case of sweetened jams, it is necessary a higher dose: 186.8 g fructose-based jam or 210.5 g stevia-based jam. The study of Jan et al. [18] has shown that stevia increases the inhibition capacity of α-glucosidase compared to other sweeteners. It has been shown that stevia used as a sweetener in food products, such as bread, tea and jams, brings the optimal amount of nutrients to the human body and also maintains blood sugar levels, preventing or controlling diabetes.

This is the first study reporting the antidiabetic action of lingonberry jams extracts exerted by inhibition of the α-glucosidase enzyme. Following in vitro study, lingonberry jams extracts exhibit notable inhibition of α-glucosidase, suggesting the presence of potential enzyme-inhibiting compounds. A possible explanation could be the presence of α-glucosidase inhibitors, such as anthocyanins, in lingonberry jams, as it was reported in a previous study [8]. Moreover, in a study reported by Wang et al. [8], no significant differences have been found between anthocyanins from *Lyciumruthenicum* fruits and acarbose in inhibiting the activity of Caco-2 cell α-glucosidase, indicating the preventing effect of anthocyanins in increasing glucose levels.

Moreover, the most abundant anthocyanin in lingonberry jams is cyanidin-3-galactoside, which was previously studied for its inhibitory effect on α-glucosidase. In a study reported by Adisakwattana et al. [19], the results showed that cyanidin-3-galactoside significantly inhibited maltase and sucrase by 35% and 64%, respectively. The mechanism through cyanidin-3-galactoside inhibiting the enzyme is still unknown, but it can be assumed that hydroxyl groups contained in the molecular structure of cyanidin-3-galactoside can form hydrogen bonds with the polar groups in the active site of protein. In this way, the enzyme molecular configuration is modified, and enzyme activity is inhibited [19]. Although cyanidin-3-galactoside was found to be less potent in inhibiting the activity of enzyme than acarbose, together they might interact synergistically on α-glucosidase. When cyanidin-3-galactoside was added to the assay system with a low dose of acarbose, the percentage inhibition was significantly increased when compared with acarbose alone [19]. Therefore, combining two inhibitors in a low dose could be more effective in the reduction of postprandial hyperglycemia. In addition, consumed in high doses, acarbose may have common gastrointestinal adverse effects [16]; thus, finding natural inhibitors that decrease the dosage of acarbose might be of relevant importance.

An important aspect that has to be taken into consideration is the absorption of anthocyanins. Although anthocyanins have been shown to be involved in α-glucosidase inhibition, they have to be firstly absorbed. The mechanism for reducing blood glucose levels depends on the anthocyanins pathway in the human body. When anthocyanins are ingested through diet and then taken into the gastrointestinal tract, they can pass the stomach wall or can reach to the small intestine, where they are degraded into other molecules, such as anthocyanidins, with an α-glucosidase inhibition effect [8]. Therefore, studying the bioavailability of anthocyanins after consumption is very important in evaluating their biological effects, such as the inhibitory action upon the α-glucosidase enzyme. The impact of in vivo digestion upon anthocyanins from lingonberry jams is discussed below.

### 2.3. Cytotoxicity and Cell Viability in Normal and CaCo-2 Cancer Cells

Cytotoxicity studies were conducted on normal dermal human fibroblast cells (BJ) and colon cancer (Caco-2) cells (Figure 4). Both normal and cancer cells were exposed to different lingonberry jams extracts at concentrations ranging between 12.5 and 200 µg/mL. Viability data show differences between normal and cancerous cells (*p* < 0.05). When normal cells are subjected to the jam’s extracts, cell viability was above the toxicity limit of 70% for all treated cells. Therefore, it is quite evident that lingonberry jams extracts show no cytotoxic effects on normal fibroblastic cells, regardless of the sweetener used in the jam formulation. The majority of lingonberry jams extracts exerted a stimulatory effect upon normal cells up to the concentration of 200 µg/mL in a dose-dependent manner.

However, the extracts of erythritol- and stevia-based jam extracts increased viability only to the concentration of 100 µg/mL. For erythritol, the maximum concentration (200 µg/mL) induced a viability decrease below the control level. Therefore, jam-treated fibroblastic cells did not show any cytotoxic effects at any of the used concentrations; moreover, they exhibited a dose-dependent stimulatory effect. Significant differences have been observed between Jam 7 and all other jams (excepting Jam 3), whereas no significant differences were noticed compared to jam without sweetener.

Colon cancer cells are more vulnerable when exposed to jams extracts than normal cells. From a statistical point of view, all jams have the same effect on cancer cells. When Caco-2 cells are subjected to jam extracts, mitochondrial activity is decreased and cell viability is lower than 70% in the case of all jams, excepting coconut sugar- and stevia-based jams. Interestingly, colon cancer cell viability is lower than 70% only at lower concentrations (12.5–25 µg/mL), excepting saccharine-based jam and unsweetened jam, where cell viability was lower than 70% in a higher range of concentrations (12.5–50 µg/mL). The strongest inhibition of cancer cell viability can be observed for saccharine-based jam extract at a 25 µg/mL concentration, where cell viability decreased to almost 40% (Figure 4). It should be noted here that Caco-2 cell viability decrease is not caused by saccharine-based extract at higher concentrations (100–200 µg/mL). These results suggest that cancer cells inhibition is dose-dependent (*p* < 0.05) and is influenced by the type of sweetener used in the jam formulation (*p* < 0.05), but no statistical dependence between these 2variables exists (*p* > 0.05). Saccharine seems to be the most suitable sweetener for the inhibition of tumor cells. Such a finding suggests that in vivo exposure to lingonberry jams digestion can damage cancer cells, with no adverse effects on normal cells. Cancer cells inhibition is dose-dependent, and statistically significant influences have been noticed in the case of Jams 1 and 3 at concentrations higher than 50 µg/mL, and in the case of Jam 7, at concentrations higher than 25 µg/mL. For Jam 7 and unsweetened jam, significant influences were observed only at higher concentrations (above 200 µg/mL). The differences between the jam extracts are possibly related to the hydroxyl groups, type and position of the sugars [20]. The combination of the anthocyanins and sweeteners also influences the stability in the medium and the cell uptake of the anthocyanins. At higher concentrations, the effect is partially stimulatory. The effect on proliferation was influenced by the amount of anthocyanins present in the extracts. At lower doses, they showed an anti-proliferative effect on CACO2 cancer cells, as previously reported [21]. At higher concentrations, the combined sugars offered an increased energetic supply and enhanced proliferation by stimulating the cells metabolic activity, which overcomes the anti-proliferative effect of the anthocyanins. Others have found that anthocyanins from fruits and vegetables, especially the strawberry and blackberry extracts, have a preventive effect in the development of colorectal cancer [21], and this effect can be more important compared to pharmacological inhibitors because of the multiple anthocyanins with a biological antitumor effect [22]. Other anthocyanins extracts showed similar effects that were reviewed by Shi et al. [21]. In CACO2 cell lines, blueberry extract inhibited proliferation, showing a greater effect comparatively with other polyphenols. China blueberry inhibited proliferation in DLD1 and COLO205 cell lines. Bilberry extract inhibited proliferation in MC38, a colon cancer line in mouse [21]. Chocoberry extract inhibited growth in the HT-29 cell line and decreased proliferation in vivo in a rat colon cancer model. Blackberry inhibited cell growth in HT-29 and CACO2 cells, and raspberry, strawberry, lingonberry, elderberry, black currant and blackthorn berries extracts showed antiproliferative effects against different colon cancer models. Grape pomace crude extracts also inhibited proliferation in CACO2 cells, but the activity was mostly attributed to the flavonoids, rather than to the anthocyanins content [23].

The inhibitory proliferation concentration 50%, IC50, was calculated for the CACO2 cell line treated with each jam extract, starting from the original OD 540 nm data, using the AAT Bioquest IC50 Calculator online tool [24]. Data are presented in Table 2.

As seen in Table 2, the anthocyanins anti-proliferative effect was strongly influenced by the combined sweetener used. Interestingly, the white sugar and the fructose showed the lowest IC50, followed by erythritol, while stevia and saccharine showed a much higher IC50. When compared to the unsweetened jam, the IC50 of the white sugar- and fructose-sweetened jams is much lower, which indicates that these sweeteners enhanced the anti-proliferative effect of the anthocyanins. Moreover, the other sweeteners, such as brown sugar and coconut sugar, inhibited the anti-proliferative effects of the anthocyanins, while the combination with stevia and saccharine sweeteners had almost no effect (Appendix A).

Literature showcasing the effects of lingonberry fruits on normal and cancer cells in vivo or in vitro is still scarce. However, the anticancer and antiproliferative effects of cranberry phytochemicals have been investigated, and extracts of these foods have demonstrated cytotoxic activity against tumor cell lines. Flavonoid-rich fraction and proanthocyanidin-rich fraction were isolated from cranberry press cake and whole cranberry, and each fraction significantly slowed the growth of explant tumors in vivo. Moreover, proanthocyanidin-rich fraction inhibited growth and induced complete regression of two other tumors explants [25]. These results indicate the potential anticancer activity of flavonoids and anthocyanins contained in cranberry extracts. In another study ongrape pomace, a major byproduct worldwide, well known for its phytochemicals, especially for anthocyanins and other phenolic compounds, the extract showed no cytotoxicity on Caco-2 intestinal cells [26]. In the study of Rugina et al. [27], in order to prove the antitumor effect of chokeberry anthocyanins, purified anthocyanin fraction containing only cyanidin glycosides (67.1% cyanidin-3-galactoside, 2.6% cyanidin-3-glucoside, 24.3% cyanidin-3-arabinoside and 5.8% cyanidin-3-xyloside) were administered to tumor cancer cells. It was found that anthocyanin fraction inhibited the survival of tumor cells by 40% at a concentration of 200 µg/mL after 48 h. In a study reported by Zhao et al. [28], a percentage of 50% of the cancer cell line was inhibited by 25 µg of cyanidin-3-glucoside/mL after a 48 h exposure to anthocyanins from chokeberry, without affecting colon epithelial cells growth. Therefore, cyanidin-based anthocyanins present in lingonberry jams extracts may be involved in the cytotoxic effect against the colon cancer cells, when used in lower concentrations.

These in vitro cytotoxicity data are only preliminary, and they need to be continued with further studies on other colon cancer models (in vitro and in vivo) and more in-depth investigations, such as different proliferation studies, cell death mechanisms and cell cycle assessment; however, the results show that even though anthocyanins present an anti-proliferative effect on colon cancer cells, as previously reported by multiple studies on in vitro and in vivo models, their activity might be strongly influenced by the other sugars, which may enhance or decrease it.

### 2.4. In Vivo Bioavailabiliy of Anthocyanins from Lingonberry Jams

For a better understanding of the action and potential protective effects of dietary anthocyanins in the human body, it is necessary to investigate their fate following ingestion. It is important to realize that anthocyanins are not necessarily the most active within the body, mostly because they are very sensitive to pH changes and poorly absorbed or rapidly eliminated [3,6]. The changes occurring in foods during the digestion process and the possible factors that may negatively influence their digestion are successfully studied nowadays using in vitro simulated digestion methods. These methods have the advantage of being fast, less expensive and easier to be applied in comparison to animal/human nutritional studies. However, due to the complexity of the digestion process, in vitro methods cannot totally reproduce the processes that occur in vivo, and further studies are needed [29]. Although studies on the content of anthocyanins and their in vitro bioaccessibility from lingonberry jams have been reported [30], the bioavailability and potential absorption of anthocyanins from these products remain unknown.

The plasma concentration profiles of the 3major anthocyanins at 1, 2, 6 and 24 h after administration are shown in Figure 5, and the concentration of each anthocyanin in the extracts of jams administered to rats are presented in Appendix A.

At only 1 h after administration, all 3peaks are detected in all plasma samples. The peak pattern is similar to that of the lingonberry jams extracts administered, indicating that anthocyanins were mainly absorbed into the body as intact. It was shown that anthocyanins can be excreted as conjugates forms [16]. The blood samples were collected from the retroorbital sinus, therefore, from the systemic circulation after passing through the liver. In the study of Talavera et al. [31], the plasma obtained from the gastric veins of rats, after in situ administration of blackberry anthocyanins, did not reveal the presence of anthocyanins metabolites, which suggests that anthocyanins are absorbed unmetabolized through the gastric wall. Moreover, Kalt et al. [32] found anthocyanins in native forms in plasma or in different tissues, such as liver, eyes, cortex or stomach, in animals. The absorption of anthocyanins is a complex process that depends on their molecular size and concentration, the characteristics of acetylation or glycosylation and the pH of the environment, as well as the interaction with digestive enzymes [33] or gut microbiota [34]. Generally, after ingestion, anthocyanins undergo a first transformation in the oral cavity, where they can be degraded by saliva enzymes and oral microbiota or can be absorbed through oral mucosa. When they reach the stomach, they are mostly absorbed due to appropriate pH. The anthocyanins that remained unabsorbed pass into the small intestine, where they are decomposed under the action of the intestinal flora, leading to the formation of metabolites, which are then taken up by the liver and distributed in different tissues and organs, where they exert biological effects. Depending on the time of their detection in the plasma and the plasmatic concentration, a correlation was established with their gastric and intestinal absorption and with the type of sweetener used. From the HPLC profile of anthocyanins, it can be concluded that the absorption of these compounds is different according to the sweetener used in the jam formulation. This fact indicates the effect of the food matrix upon anthocyanins bioavailability. The results show that all 3anthocyanins are absorbed following administration and are still detectable in the bloodstream after 24 h. This fact suggests that anthocyanins from lingonberry jams have a prolonged bioavailability in rats.

In the case of erythritol-, white sugar-, stevia- and saccharine-based jam, anthocyanins are highly released in the plasma after only 1–2h after administration, indicating that they are rapidly absorbed (Figure 5) and can permeate the gastric barrier [35]. In the case of fructose- and brown sugar-based jams, anthocyanins reach their maximum level at 6 h after ingestion, indicating that these sweeteners could slow the absorption of anthocyanins. Interestingly, after administration of coconut sugar-based jam, all 3anthocyanins appear in a high concentration in the bloodstream after only 1 h, maintain a plateau and then reach a maximum concentration at 24 h. In coconut sugar-based jam, the maximum concentration of the most abundant anthocyanin, cyanidin-3-galactoside was 56.1 µM, which is 30.8% of the intake. The levels of anthocyanins released in plasma depend on the food matrix, absorption through the gastrointestinal wall and excretion to urine and bile [35]. In a study reported by Ichiyanagi et al. [35], anthocyanins were released at only 15 min after oral administration of bilberry extract and then decreased with time. On the other hand, a study on lingonberry fruits [36] reports that anthocyanins were absorbed more slowly, and the peak concentrations in urine remained lower than in previous anthocyanin absorption studies. This indicates that anthocyanins absorption is different depending on the provenience source and interaction with other compounds, such as sweeteners used in our study that may accelerate or slow their maximum absorption into the bloodstream. There are studies that strongly indicate that food matrix may increase anthocyanin absorption, and thus, bioavailability of these molecules may be improved by different coexisting food components [35]. Moreover, their interaction with a food matrix, especially proline-richproteins, can increase the stability of anthocyanins in terms of color, thermal stability and resistance to oxidative attacks and prevent their degradation during gastrointestinal transit. Other studies stated that different sweeteners can influence the plasma concentration of polyphenolics due to the effect on their pharmacokinetics [37]. It seems that low-weight polysaccharides, including sucrose, can increase the intestinal absorption of phenolic compounds by enhancing the motility and secretion or due to activation of membrane transporters. However, it is not known exactly how much is due to the interaction with the food matrix or how much is related to the stimulation of digestive processes [38]. The bioavailability of anthocyanins was also calculated, and the results are given in Table 3.

The calculated bioavailability ranges from 4.85 to 32.7%. The highest bioavailability of anthocyanins is observed in the presence of coconut sugar, whereas the lowest bioavailability was determined in brown sugar-based jam. In this study, the three main anthocyanins found in lingonberry jams have the same aglycone, but a different sugar moiety structure: cyanidin-3-galactoside, cyanidin-3-glucoside and cyanidin-3-arabinoside. When the bioavailability of anthocyanins with a different sugar moiety is compared, cyanidin-3-glucoside tends to show the highest bioavailability in the case of Jams 2, 4, 5 and 7. For Jams 1, 2 and 3, cyanidin-3-arabinoside is the most bioavailable. It was reported that anthocyanins bioavailability depends on the sugar moiety attached, galactosides being better absorbed than the other sugar moiety [35]. Remarkable, cyanidin-3-galactoside shows quite lower bioavailability values than the other two anthocyanins in the case of the majority of jams. Thus, the bioavailability of these 3anthocyanins is influenced (*p* < 0.05) by the different sweeteners used in the jam formulation. However, anthocyanins bioavailability is also influenced by other factors, such as the metabolism or hydrophobicity of the aglycone B ring [35]. The obtained bioavailability values of anthocyanins from lingonberry jams are much higher than those reported by Ichiyanagi et al. [35] in a study on bilberry fruits, where the bioavailability ranged from 0.61 to 1.82%. This fact reinforces the idea that plasma concentrations vary highly according to the nature of the anthocyanins and the food source. Although any differences in the bioavailability of the three anthocyanins are not statistically significant, statistically significant differences were noticed between jams. For example, the bioavailability of both cyanidin-3-galactoside and cyanidin-3-arabinoside in Jams 1, 2, 4 and 7 is statistically significantly different from that in Jams 5 and 6, whereas cyanidin-3-glucoside and cyanidin-3-arabinoside bioavailability in Jam 3 statistically significantly differs from that in Jam 5.

The limits of the in vivo study are related to the small number of animals used for the study, the lack of correlation between the content of anthocyanins in serum and different tissues (liver, kidney, gut, heart) and the endogenous antioxidant defense evaluated by the DPPH and ABTS tests or by antioxidant enzymes activity from erythrocyte lysates (superoxide dismutase, catalase, glutathione peroxidase). It is important to perform these studies on healthy animals and also on animals with diabetes or obesity, in correlation with different pathways triggered by anthocyanins or their metabolites (inflammation, apoptosis, cell migration, redox imbalance).

## 3. Materials and Methods

### 3.1. Chemicals

Ethanol 96%, citric acid, acetic acid, trichloroacetic acid (TCA), formic acid, acetonitrile (ACN), sodium phosphate salts (Na_2_HPO_4_, NaH_2_PO_4_), p-nitrophenyl-α-D-glucopyranoside (p-NPG), α-glucosidase from rice, sodium carbonate, acarbose and cyaniding were purchased from Merck (Darmstadt, Germany). All reagents used in the experimental part were of analytical purity.

### 3.2. Lingonberry Jams and Extracts Preparation

Jams were prepared according to our previous study [16], with slight modifications. Lingonberry jams were prepared without additives and using different sweeteners: sucrose (Jam 1), fructose (Jam 2), erythritol (Jam 3), brown sugar (Jam 4), coconut sugar (Jam 5), stevia (Jam 6), saccharine (Jam 7). Lingonberry fruits were first grounded and then filtered through a sieve, in order to obtain a very fine texture that could be easily administrated to rats. Each jam was prepared using 100 g of fruit and following quantities of sweeteners established taking into account the sweetening degree of each sweetener: 50 g sucrose (Jam 1), 29.4 g fructose (Jam 2), 77.0 g erythritol (Jam 3), 50.0 g brown sugar (Jam 4), 50.0 g coconut sugar (Jam 5), 0.180 g stevia (Jam 6) and 0.180 g saccharine (Jam 7).The heating temperature was 50 °C in order to avoid anthocyanins degradation. The jams were considered finished when the value of the total soluble content (TSS) reached a value of 56–57° Brix, less in the case of Jams 6 and 7, respectively, when the heating was stopped when the TSS reached approximately 22° Brix [16,30]. Jams were not pasteurized. After cooling to room temperature, jams were divided in order to analyze the effects of sweeteners upon anthocyanins stability under 3different storage conditions during the 180 days: 4 °C (refrigerator), 25 °C (light) and 25 °C (dark).

The jam extracts were prepared by mixing 1.0 g of each lingonberry jam with 4.5 mL ethanol and 5.5 mL ultrapure water. The mixture was magnetically stirred for 60 min at room temperature. An ultrasonic thermostatic bath Elmasonic E60H (Elma Schmidbauer GmbH, Singen am Hohentwiel, Germany) was used for extraction of target compounds. Ultrasound-assisted extraction was performed at 30 °C for 30 min. The extracts were centrifuged at 875× *g* for 20 min using a Centurion Scientific centrifuge C2006 (Centurion Scientific Limited, Bosham, UK). Each extract was filtered out, and the supernatants were collected and used for further analyses.

Both the jams and their extracts were freshly prepared before each test.

### 3.3. High-Performance Liquid Chromatography (HPLC) Analysis of Anthocyanins

#### 3.3.1. Lingonberry Jams Extracts Analysis

For the analysis of anthocyanins from lingonberry jams extracts, the method of Zheng et al. [13] was applied, with some modifications. Lingonberry jams samples were passed through 0.2 µm membrane filters. A volume of 20 µL was injected into an Agilent 1200 HPLC system (Agilent Technologies Inc.; Santa Clara, CA, USA) equipped with a diode array detector. An Eclipse XTB-C18 (Agilent) column (150 × 4.6 mm inner diameter, particle size 5 µm) was used as stationary phase. There were 2solvents, 5% aqueous formic acid (A) and HPLC grade acetonitrile (B) composed the mobile phase. A gradient profile was used in this experimental part: 0–1 min, 4% B; 1–10 min, 4–6% B; 10–15 min, 6% B; 15–35 min, 6–18% B. The flow rate was 1 mL/min, and the detection was performed at 532 nm.

#### 3.3.2. HPLC Quantification of Anthocyanins in Plasma

The anthocyanins from plasma samples were analyzed by the HPLC method described by Kostka et al. [14], with some minor modification. The previously described Agilent 1200 HPLC system, including the Eclipse XTB-C18 column, was used. The mobile phase consisted of water/acetonitrile/formic acid (95/3/2; *v*/*v*/*v*) (eluent A) and water/acetonitrile/formic acid (48/50/2; *v*/*v*/*v*) (eluent B), the gradient profile being the following: start—6% eluent B; 0–30 min, 35% B; 30–35 min, 40% B; 35–45 min, 90% B; 45–50 min 90% B; 50–55 min, 30% B; 55–70 min, 6% B. The flow rate was 200 μL/min, and the quantification was performed at λ = 520 nm using cyanidin as the internal standard compound. Aliquots of 20 μL of each spiked plasma samples (3:1 *v*/*v* with jam extract) were analyzed in triplicate.

### 3.4. α-Glucosidase Inhibition Assay

The α-glucosidase inhibition assay was adapted from Mohamed Sham Shihabudeen et al. [9]. The enzyme solution (1 U/mL) was prepared by adding 990 µL of phosphate buffer (50 Mm; pH 6.9) to 10 µL of α-glucosidase. A reaction mixture containing 200 µL of phosphate buffer (50 Mm; pH 6.9), 300 µL α-glucosidase (1 U/mL) and 200 μL of lingonberry jam extract was preincubated for 5 min at 37 °C, and then 200 μL of 20 mM p-NPG was added to the mixture as a substrate. After further incubation at 37 °C for 60 min, the reaction was stopped by adding 50μLof Na_2_CO_3_ (0.1 M). Thereafter, the absorbance was measured at 405 nm, using a T80+ UV-Vis spectrophotometer (PG Instruments, Lutterworth, UK). Acarbose was used as a positive control. The blank sample was prepared using all components except α-glucosidase, which was replaced with the same volume of 50 mM sodium phosphate buffer. The percentage of enzyme inhibition by the sample was calculated by the following equation:% Inhibition = {[(AC − AS)/AC] × 100},(1)
where AC is the absorbance of the control, and AS is the absorbance of the tested sample. All experiments were conducted in triplicate.

### 3.5. Cytotoxicity Assay

#### 3.5.1. Cell Cultures

The assessment was performed on a cancer cell line of Caucasian colon adenocarcinoma Caco-2 (ECCAC, Sigma Aldrich, Co Heidelberg, Germany), and respectively dermal human normal fibroblasts (BJ, ATCC CRL-2522™, Manassas, VA, USA). Cells were cultured in Dulbecco modified Eagle medium (DMEM) supplemented with 5% fetal calf serum, 50 µg/mL gentamicin and 5 ng/mL amphotericin, all from Biochrome AG (Berlin, Germany), in standard cell culture conditions, 5%CO_2_ and 37 °C high humidity. The medium was changed twice weekly.

#### 3.5.2. Cytotoxicity Assay

Cytotoxicity testing employed two cell lines, on which the protocol proposed by Filip et al. [39], with minor modification, was applied. Cells, either Caco-2 or BJ, were cultivated in ELISA 96 wells micro titration flat-bottom plaques (TPP, Trasadingen, Switzerland) at a density of 10^4^/well and settled for 24 h. Cells were then treated for 24 h with different concentrations of each jam extract. The jam extracts were prepared in phosphate saline buffer (PBS) immediately before use, then diluted with fresh medium to reach final concentrations of 12.5, 25, 50, 100 and 200 µg/mL. Cytotoxicity was evaluated using the CellTiter 96^®^AQueous Non-Radioactive Cell Proliferation Assay (Promega Corporation, Madison, WI, USA), as indicated by the producer. Readings were conducted using an ELISA plate reader at 540 nm (Tecan, Männedorf, Switzerland). Untreated cultures exposed to medium were used as controls. Experiments were performed in triplicate. Cytotoxicity is presented as % of untreated controls; the toxicity limit was considered 70%. The inhibitory concentration 50% of proliferation (IC50) was calculated for the CACO2 cell line treated with each jam extract, starting from the original OD 540 nm data, using the AAT Bioquest IC50 Calculator online tool [24].

### 3.6. Animal Study Design and Plasma Analysis

The applied animal study design followed the protocol of Ichiyanagi et al. [35], with minor adjustments. In total, 79 male Wistar rats weighing ~180 g were housed in cages, in a 12 h light–12 h dark cycle, at room laboratory conditions (temperature 24 ± 2 °C and humidity 65%) and were fed with standard food (autoclavable rodent diet VFR1) and water ad libitum. All experimental procedures complied with ARRIVE guidelines and were approved by the Animal Ethics Board of “Iuliu Hatieganu” University on animal welfare according to the Directive 2010/63/EU on the protection of animals used for scientific purposes (No. 261/20 May 2021). The rats were deprived of food, but had access to water 16 h before and 24 h after experiment. The animals were divided into 2 groups: control group (control rats, n = 8) and group fed with lingonberry jams (lingonberry jams fed rats, n = 71). The control group was fed with 1 mL citric acid (0.1%), whereas the second group was fed with 1 mL of lingonberry jam extracts. Lingonberry jam extracts were prepared by vortexing4 mL of each jam with 0.5 mL citric acid 0.1%. The 1 mL dose of lingonberry jam extract was orally administered to experimental animals by gavage. Control rats were given the same volume of 0.1% citric acid. Blood (3 mL) was withdrawn from the retroorbital sinus on EDTA at 1 h, 2 h, 6 h and 24 h and then centrifuged for 10 min and acidified by the addition of 10 μL of acetic acid 1 M to 1 mL of plasma, as proposed in the study of Talavera et al. [40]. All samples were rapidly frozen and stored at −20 °C until analysis. Plasma analysis was performed according to Felgines et al. [41]. Plasma was treated 2:1 with 1.2 mol/L trichloroacetic acid to precipitate proteins and then centrifuged for 5 min at 12,000× *g* at room temperature. The supernatant was further analyzed by the HPLC method.

Taking into account that the bioavailability index is calculated as the ratio between the amount of compound in the absorbed fraction and the amount of compound in the non-digested sample, the bioavailability (%) of anthocyanins after 1 h, 2 h, 6 h and 24 h was calculated according to the formula used for in vitro digestion, as well as [42]:Bioavailability (%) = C_p_/C_i_ × 100,(2)
where C_p_—concentration in plasma at a given time (h) and C_i_—initial concentration.

### 3.7. Statistical Analysis

All experiments were performed in triplicate, and the obtained data were statistically analyzed by means of I-test and analysis of variance (ANOVA). The Software Package Statistica 12.0 (StatSoftInc. 1984–2013, Tulsa, OK, USA) was used, considering the confidence level at 95%, for which the significance rank is *p* < 0.05.

## 4. Conclusions

As far as we know, this is the first research describing the influence of different sweeteners upon the stability and in vivo bioavailability of anthocyanins from lingonberry jams. Moreover, this study brings important knowledge regarding the interaction of two human cell lines with lingonberry jams extracts, as well as the impact of these extracts uponα-glucosidase enzyme. Anthocyanins were significantly decreased as a result of jam storage, and this brings challenges to the food industry. Coconut sugar negatively influenced the stability of anthocyanins during storage. On the contrary, fructose and stevia showed a protective effect upon anthocyanins during storage, indicating their potential use in functional foods development. The bioavailability and absorption of anthocyanins was mainly governed by the sweeteners used in the jam formulation. Glucosides were absorbed more effectively than galactosides in the majority of cases. The highest bioavailability of anthocyanins was observed in the presence of coconut sugar. Fructose- and stevia-based jams were also found to possess the highest inhibition activity upon the α-glucosidase enzyme (up to 43%), indicating their benefits in preventing or controlling diabetes. Colon cancer cells were more vulnerable when exposed to jams extracts than normal cells. Cancer cells inhibition was highly influenced by the administered dose of extract and by the sweetener used. From a statistical point of view, all jams had the same effect on cancer cells. The results obtained in this research will help to develop new functional foods.

## Figures and Tables

**Figure 1 foods-12-02569-f001:**
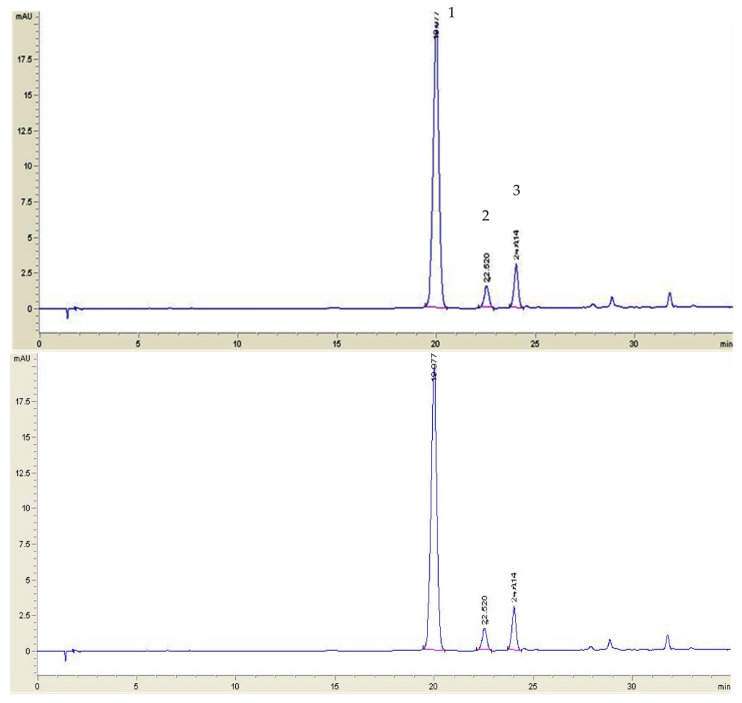
HPLC profile (532 nm) of anthocyanins in lingonberry jams: (1) cyanidin-3-galactoside; (2) cyanidin-3-glucoside; (3) cyanidin-3-arabinoside.

**Figure 2 foods-12-02569-f002:**
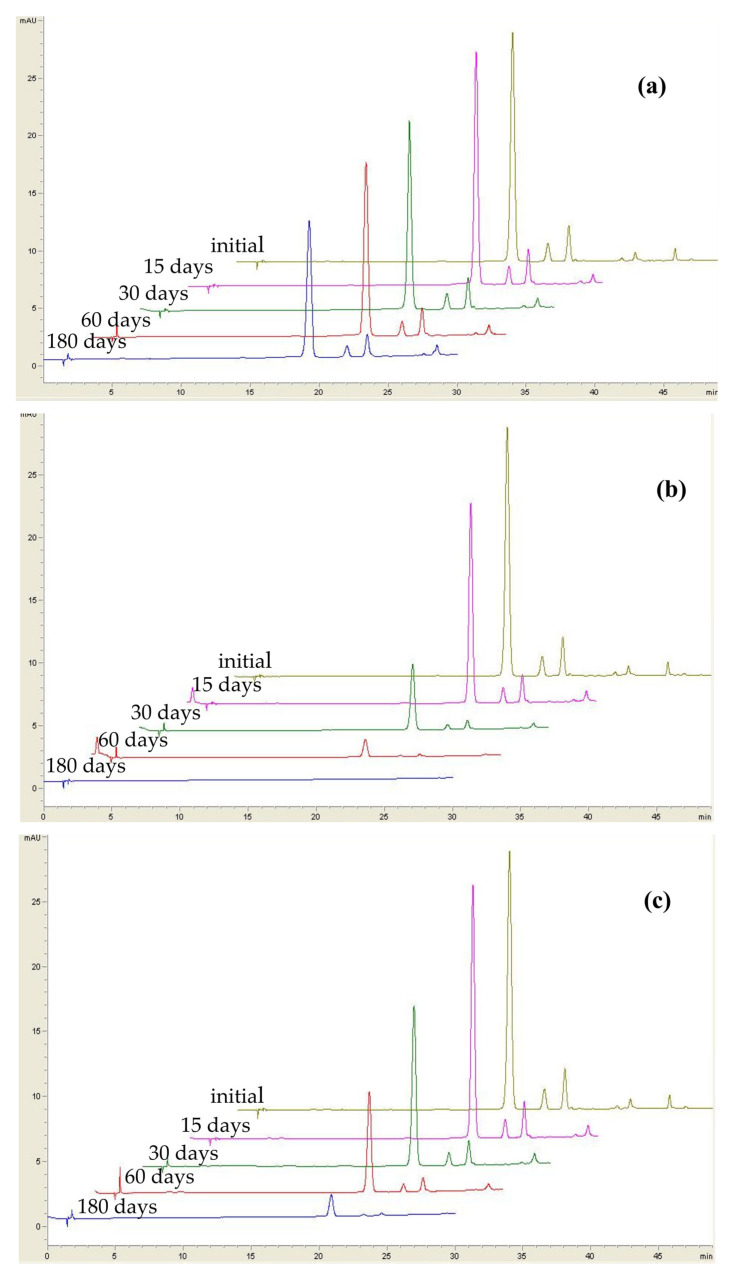
Changes in HPLC profile of individual anthocyanins from fructose-based jam under different storage conditions: (**a**)refrigeration; (**b**) light; (**c**) dark.

**Figure 3 foods-12-02569-f003:**
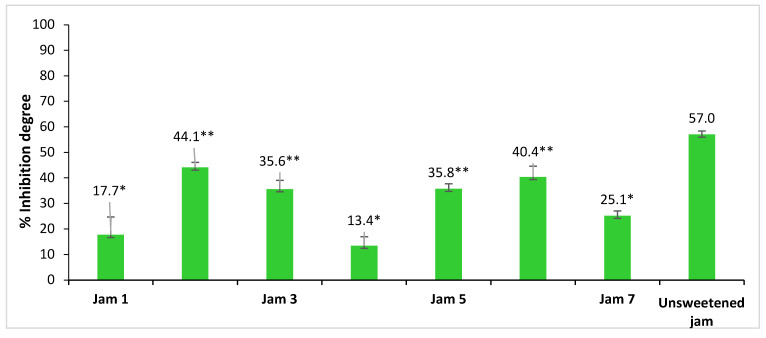
The percentage enzyme inhibition of lingonberry jams. The sweeteners used in jams were Jam 1—sucrose, Jam 2—fructose, Jam 3—erythritol, Jam 4—brown sugar, Jam 5—coconut sugar, Jam 6—stevia, Jam 7—saccharine. Asterisk symbols signify the following levels of statistical significance of differences between jams with sweeteners and unsweetened jam according to one way ANOVA: ** *p* < 0.0001, * *p* < 0.001.

**Figure 4 foods-12-02569-f004:**
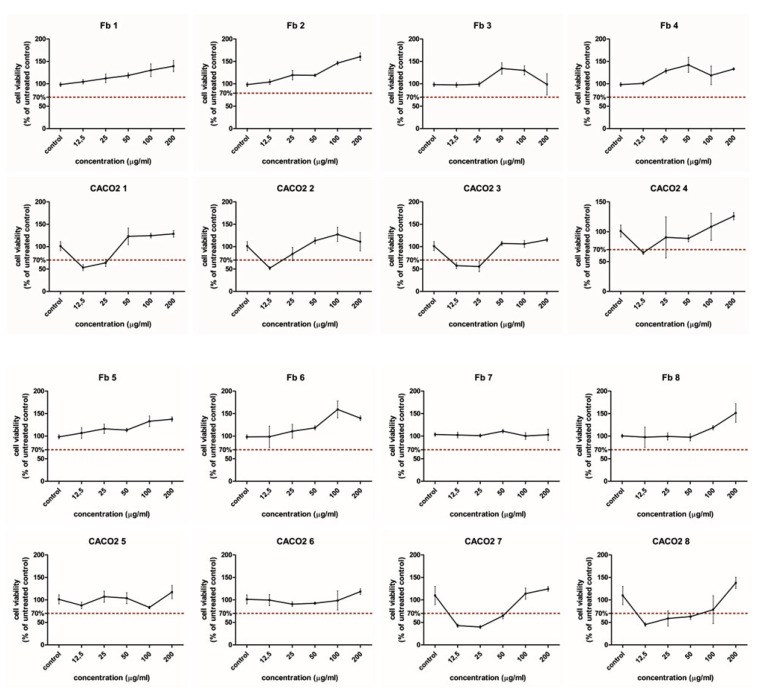
Comparative cell viability in the normal fibroblast cell cultures and colon cancer CACO2 cell line treated with each jam extract in concentrations ranging from 0 to 200 µg/mL; upper panels—fibroblasts, lower panels—CACO2 cells. Data are presented as % of the untreated controls, for each experiment mean ± SD, n = 3 is presented. Jam 1 = white sugar, Jam 2 = fructose, Jam 3 = erythritol, Jam 4 = brown sugar, Jam 5 = coconut sugar, Jam 6 = stevia, Jam 7 = saccharine, Jam 8 = unsweetened jam.

**Figure 5 foods-12-02569-f005:**
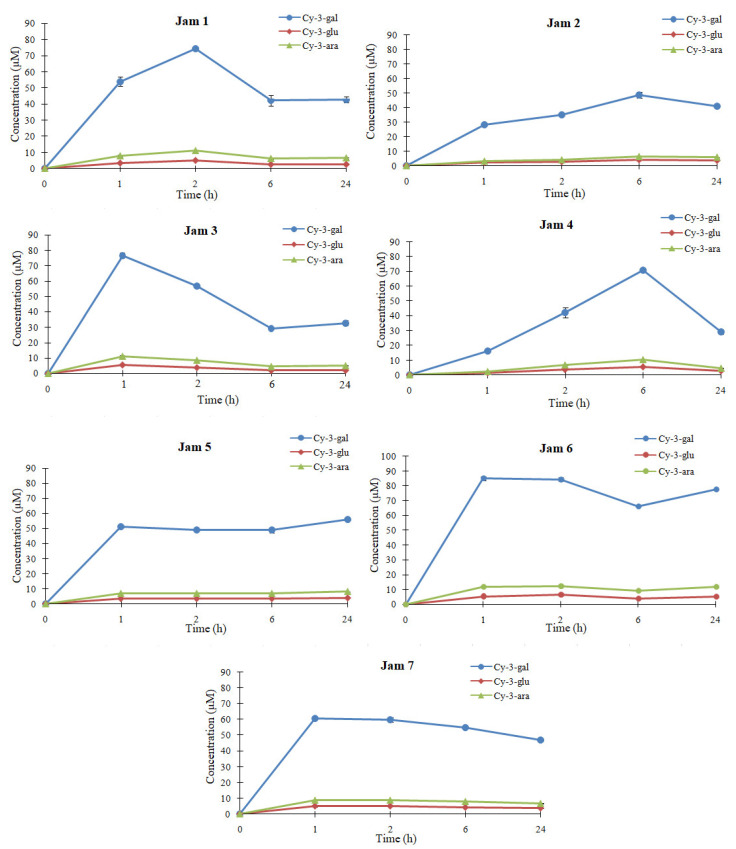
Plasma anthocyanin concentration profiles in rats after administration of lingonberry jams. Cy-3-gal: cyanidin-3-galactoside; Cy-3-glu: cyanidin-3-glucoside; Cy-3-ara: cyanidin-3-arabinoside. The sweeteners used in jams were Jam 1—sucrose, Jam 2—fructose, Jam 3—erythritol, Jam 4—brown sugar, Jam 5—coconut sugar, Jam 6—stevia, Jam 7—saccharine. Data are presented as mean ± SD (n = 3).

**Table 1 foods-12-02569-t001:** Changes in individual anthocyanin content in lingonberry jams during 180 days of storage at 4 °C, 25 °C (light conditions) and 25 °C (dark conditions), expressed as percentages (%). Data values determined in initial jams (day 0) are set as 100%.

Sample	Anthocyanin	4 °C				25 °C (Light Conditions)				25 °C (Dark Conditions)			
		Day 15	Day 30	Day 60	Day 180	Day 15	Day 30	Day 60	Day 180	Day 15	Day 30	Day 60	Day 180
Jam 1 (white sugar)	Cyd-3-gal	91.1 ± 3.2	88.7 ± 4.1	77.9 ± 2.2	61.9 ± 2.8	71.2 ± 3.4	26.2 ± 1.5	7.10 ± 0.41	0.10 ± 0.01	83.9 ± 4.4	61.2 ± 3.3	37.7 ± 1.1	8.01 ± 0.44
	Cyd-3-glu	94.5 ± 2.9	89.1 ± 3.9	86.7 ± 3.1	59.3 ± 3.4	72.6 ± 4.2	25.9 ± 1.8	6.31 ± 0.22	0.51 ± 0.03	94.1 ± 5.8	64.4±	33.8 ± 2.3	5.90 ± 0.21
	Cyd-3-ara	96.9 ± 4.1	88.6 ± 3.5	81.2 ± 2.8	63.1 ± 2.6	72.8 ± 2.5	19.9 ± 0.9	5.90 ± 0.19	0.32 ± 0.01	89.7 ± 4.7	59.1±	32.5 ± 1.9	4.81 ± 0.14
Jam 2 (fructose)	Cyd-3-gal	95.1 ± 3.3	94.2 ± 5.2	86.7 ± 2.6	67.6 ± 1.9	60.4 ± 2.9	26.3 ± 1.6	10.3 ± 0.8	0.11 ± 0.01	90.7 ± 6.1	54.0±	26.2 ± 1.0	1.42 ± 0.08
	Cyd-3-glu	99.2 ± 5.0	90.9 ± 4.6	87.6 ± 3.2	62.9 ± 2.2	60.4 ± 1.8	24.2 ± 1.0	10.8 ± 0.5	n.d.	97.2 ± 5.9	50.8±	24.2 ± 1.5	0.66 ± 0.00
	Cyd-3-ara	95.2 ± 1.9	90.4 ± 3.9	82.5 ± 3.3	68.6 ± 3.4	55.5 ± 1.2	18.9 ± 1.0	10.7 ± 0.3	0.30 ± 0.01	96.7 ± 5.1	50.3±	25.8 ± 1.1	0.72 ± 0.02
Jam 3 (erythritol)	Cyd-3-gal	93.5 ± 2.1	80.9 ± 2.1	78.2 ± 2.9	64.5 ± 3.8	38.9 ± 3.3	14.7 ± 0.8	4.70 ± 0.14	n.d.	73.9 ± 4.0	61.9±	38.2 ± 2.3	n.d.
	Cyd-3-glu	96.3 ± 4.2	87.5 ± 1.7	83.3 ± 3.6	60.8 ± 2.8	46.6 ± 1.8	14.2 ± 0.7	5.83 ± 0.25	n.d.	74.4 ± 4.4	69.2±	34.1 ± 2.0	n.d.
	Cyd-3-ara	92.8 ± 2.3	81.3 ± 3.0	79.9 ± 4.1	66.9 ± 1.9	35.6 ± 2.1	9.4 ± 0.62	4.54 ± 0.33	n.d.	74.9 ± 3.8	61.2±	32.6 ± 1.9	n.d.
Jam 4 (brown sugar)	Cyd-3-gal	91.3 ± 4.9	89.7 ± 2.8	88.3 ± 2.2	61.3 ± 2.4	42.2 ± 2.0	18.5 ± 1.0	8.01 ± 0.41	0.32 ± 0.02	68.5 ± 2.3	56.9±	31.7 ± 2.4	4.02 ± 0.11
	Cyd-3-glu	96.7 ± 5.3	93.1 ± 4.0	83.6 ± 3.7	60.9 ± 3.3	38.7 ± 1.7	18.7 ± 1.3	8.83 ± 0.28	n.d.	59.9 ± 1.9	51.1±	29.2 ± 1.8	1.11 ± 0.08
	Cyd-3-ara	85.7 ± 1.8	82.9 ± 3.5	80.4 ± 4.5	61.8 ± 2.9	30.9 ± 2.1	15.9 ± 1.7	6.71 ± 0.33	n.d.	62.2 ± 3.0	55.4±	31.4 ± 2.0	2.23 ± 0.06
Jam 5 (coconut sugar)	Cyd-3-gal	97.7 ± 3.9	67.9 ± 2.8	66.4 ± 2.9	20.5 ± 1.0	31.4 ± 1.6	7.80 ± 0.61	1.92 ± 0.09	0.10 ± 0.01	40.5 ± 1.5	16.4±	5.21 ± 0.23	0.33 ± 0.00
	Cyd-3-glu	75.6 ± 1.8	64.9 ± 1.9	51.6 ± 1.6	15.3 ± 1.1	27.1 ± 1.9	5.40 ± 0.31	7.72 ± 0.37	n.d.	29.8 ± 1.1	16.1±	4.23 ± 0.15	0.52 ± 0.01
	Cyd-3-ara	99.1 ± 6.1	72.1 ± 2.4	71.3 ± 2.7	19.6 ± 1.6	28.8 ± 1.1	10.4 ± 0.82	2.52 ± 0.11	n.d.	40.4 ± 3.6	20.8±	3.80 ± 0.12	0.43 ± 0.02
Jam 6 (stevia)	Cyd-3-gal	98.1 ± 4.8	97.3 ± 5.3	90.4 ± 1.1	77.5 ± 2.8	46.1 ± 3.1	12.8 ± 0.64	2.72 ± 0.15	n.d.	82.9 ± 5.8	52.7±	52.1 ± 3.6	n.d.
	Cyd-3-glu	97.2 ± 3.3	98.2 ± 6.0	97.5 ± 5.6	51.6 ± 1.7	46.5 ± 3.0	12.1 ± 0.53	6.22 ± 0.39	n.d.	80.4 ± 4.7	54.6±	38.6 ± 1.0	n.d.
	Cyd-3-ara	99.2 ± 6.0	93.6 ± 4.8	87.5 ± 4.0	55.8 ± 1.0	37.8 ± 2.8	9.60 ± 0.22	4.42 ± 0.30	n.d.	85.7 ± 3.0	51.9±	42.6 ± 3.1	n.d.
Jam 7 (saccharine)	Cyd-3-gal	93.1 ± 4.4	91.8 ± 2.7	83.1 ± 3.8	66.9 ± 2.3	31.8 ± 2.4	7.40 ± 0.41	1.72 ± 0.10	n.d.	76.1 ± 2.1	46.2±	40.7 ± 2.7	n.d.
	Cyd-3-glu	92.7 ± 2.2	91.3 ± 4.1	72.6 ± 2.7	62.8 ± 2.8	32.4 ± 2.2	6.81 ± 0.43	4.30 ± 0.23	n.d.	73.8 ± 0.2.2	41.5±	25.8 ± 1.1	n.d.
	Cyd-3-ara	95.7 ± 3.7	94.2 ± 3.6	85.9 ± 2.2	68.5 ± 3.7	20.5 ± 1.1	5.31 ± 0.35	3.51 ± 0.09	n.d.	78.5 ± 3.4	44.4±	33.1 ± 2.1	n.d.

n.d.—not detected. Values represent mean ± SD (n = 3).

**Table 2 foods-12-02569-t002:** Comparative inhibitory concentration of proliferation 50% (IC50) in the colon cancer CACO2 cell line treated with the jam extracts. Calculation was performed from the original cell cytotoxicity data, using the AAT Bioquest IC50 calculator on line.

Jam	Sweetener	IC50 (µg/mL)
1	White sugar	35.51
2	Fructose	35.52
3	Erythritol	44.60
4	Brown sugar	202.0
5	Coconut sugar	230.9
6	Stevia	92.29
7	Saccharine	90.87
8	Unsweetened	117.3

**Table 3 foods-12-02569-t003:** Bioavailability of anthocyanins in the plasma of rats 1–24 h after the administration of lingonberry jams.

Anthocyanins	Bioavailability (%)						
	Jam 1 (White Sugar)	Jam 2 (Fructose)	Jam 3 (Erythritol)	Jam 4 (Brown Sugar)	Jam 5 (Coconut Sugar)	Jam 6 (Stevia)	Jam 7 (Saccharine)
**1 h**							
Cy-3-gal	15.2 ± 0.9 ^a^	8.10 ± 0.3 ^a,b^	28.3 ±1.0 ^a,b,c^	4.91 ± 0.3 ^a,b,c,d^	28.1 ± 0.2 ^a,b,d,e^	28.0 ± 0.7 ^a,b,f^	20.3 ± 0.4 ^a,b,c,d,e,f^
Cy-3-glu	12.4 ± 1.9 ^a^	8.21 ± 0.9 ^b^	26.1 ±0.6 ^a,b,c^	5.30 ± 0.1 ^c,d^	29.1 ± 1.5 ^a,b,d,e^	23.8 ± 0.1 ^a,b,c,d,^	22.6 ± 1.4 ^a,b,d,e^
Cy-3-ara	15.5 ± 0.3 ^a^	7.11 ± 0.2 ^a,b^	27.9 ±1.3 ^a,b,c^	5.11 ± 0.7 ^a,c,d^	28.5 ± 0.3 ^a,b,d,e^	29.1 ± 1.1 ^a,b,d,f^	20.6 ± 0.1 ^a,b,c,d,e,f^
**2 h**							
Cy-3-gal	20.9 ± 0.1 ^a^	10.0 ± 0.1 ^a,b^	20.9 ±0.1 ^b,c^	12.7 ± 1.2 ^a,c,d^	27.0 ± 0.7 ^a,b,c,d,e^	27.8 ± 0.7 ^a,b,c,d,f^	20.0 ± 0.6 ^b,d,e,f^
Cy-3-glu	18.0 ± 1.3 ^a^	10.2 ± 0.0 ^a,b^	17.8 ±1.4 ^b,c^	14.9 ± 0.5 ^b,d^	28.0 ± 1.4 ^a,b,c,d^	28.7 ± 1.2 ^a,b,c,d,e^	22.8 ± 1.4 ^b,d,e^
Cy-3-ara	21.6 ± 0.5 ^a^	9.01 ± 0.4 ^a,b^	21.7 ±0.5 ^b,c^	14.4 ± 0.6 ^a,b,c,d^	28.0 ± 1.4 ^a,b,c,d,e^	29.7 ± 0.4 ^a,b,c,d,f^	20.3 ± 0.6 ^b,d,e,f^
**6 h**							
Cy-3-gal	11.9 ± 1.0 ^a^	13.8 ± 0.8 ^b^	10.7 ±0.3 ^b,c^	21.4 ± 0.7 ^a,b,c,d^	26.9 ± 1.6 ^a,b,c,d,e^	21.8 ± 0.4 ^a,b,c,e,f^	18.3 ± 0.1 ^a,b,c,d,e,f^
Cy-3-glu	9.11 ± 1.7 ^a^	15.3 ± 0.2 ^a,b^	9.28 ±0.4 ^b,c^	23.0 ± 0.6 ^a,b,d^	27.3 ± 1.8 ^a,b,c,e^	16.9 ± 0.1 ^a,b,c,d,e,f^	20.3 ± 0.5 ^a,b,d,e,f^
Cy-3-ara	12.1 ± 0.1 ^a^	13.1 ± 0.1 ^a,b^	11.5 ±0.8 ^c^	22.3 ± 0.5 ^a,b,c,d^	28.0 ± 1.4 ^a,b,c,d,e^	22.4 ± 0.7 ^a,b,c,e,f^	18.5 ± 0.1 ^a,b,c,d,e,f^
**24 h**							
Cy-3-gal	12.1 ± 0.6 ^a^	11.6 ± 0.4 ^b^	12.1 ±0.8 ^b,c^	8.81 ± 0.6 ^a,b,c,d^	30.8 ± 0.9 ^a,b,c,d,e^	25.6 ± 0.3 ^a,b,c,d,e,f^	15.7 ± 0.1 ^a,b,d,e,f^
Cy-3-glu	9.80 ± 1.6 ^a^	14.6 ± 0.1 ^a,b^	9.71 ±0.3 ^b,c^	10.9 ± 0.2 ^b,d^	32.0 ± 0.6 ^a,b,c,d,e^	22.8 ± 0.3 ^a,b,c,d,e,f^	17.1 ± 0.7 ^a,b,d,e,f^
Cy-3-ara	12.9 ± 0.6 ^a^	12.0 ± 0.3 ^b^	12.8 ±0.9 ^c^	9.90 ± 0.7 ^d^	32.7 ± 0.4 ^a,b,c,d,e^	28.8 ± 0.4 ^a,b,c,d,e,f^	15.7 ± 0.1 ^a,b,c,d,e,f^

Values represent mean ± SD (n = 3). Statistically significant differences (*p* < 0.05) according to Student’s *t*-tests are indicated by letters ^a,b,c,d,e^ and ^f^.

## Data Availability

The data presented in this study are available in this manuscript.

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
