# Peer review of "Sweeteners’ Influence on In Vitro α-Glucosidase Inhibitory Activity, Cytotoxicity, Stability and In Vivo Bioavailability of the Anthocyanins from Lingonberry Jams"

_foods, 2023, doi:10.3390/foods12132569_

Round 1

Reviewer 1 Report

as attached file

Author Response

In academic papers, it is generally recommended to use a declarative title that clearly and succinctly conveys the topic or main focus of the paper. While a question title may be intriguing, it is often more appropriate for other forms of writing, such as blog posts or magazine articles.

R: The title was.

Align the axis

R: Done

Jam 2, jam 4, jam 6

R: Included. The original manuscript has these notes. I don't know why you don't see it in your manuscript

check the letters , How do you compare the difference data? Compare between row or column?

R: I checked the letters and I made corresponding changes. The comparison was done with the data from the rows.

Temperature

R: 370C. Inserted in the text

How much?

R: Medium was changed twice weekly, according to the cell culture flask used for multiplication of the cells. Manuscript was modified accordingly

Why chhose this condition

R: Because the VRF1 diet can be consumed ad libitum with fresh water

Give the name

R: The name of food is autoclavable rodent diet VFR1. I mention in the text

Check format

R: The format was modified accordingly

Reviewer 2 Report

The study investigated variations of anthocyanins in lingonberry jams. Effect of storage conditions on the contents of anthocyanins were analyzed in jams prepared with different sweeteners. The study can be beneficial to consumers, but it needs to be improved for scientific reasons.

1.   Did you measure anthocyanins contents in lingonberry before making jams? Then should be included.

2.  Stevia and saccharine are alternative sweetener that have much stronger sweetness than conventional sugar. Would you provide a recipe for each of jam? The real recipe could affect the state of anthocyanins in the jams.

3. Tables, you should provide statistical analysis results. 

4.  Provide standard curve of acarbose inhibition effects on the a-glucosidase. By the way, what kind of a-glucosidase did you use? 

5. Clearly indicate jams in what storage conditions were used in each in vitro and vivo experiments.

6. In cell viability assay, jam extracts, 1, 7 and 8, did not show dose dependent behavior for Caco-2 cells. How could you explain this? It is difficult to say this as inhibition by the extract.

7. In PK study, I could not believe the std deviations in each group.  Check the data again. 

8. Methods descriptions are unacceptable quality.

There must be some improvement on the descriptions.

Author Response

  1.  Did you measure anthocyanins contents in lingonberry before making jams? Then should be included.

R: Although the total anthocyanin content of lingonberry was determined before making jams, I don't think it should be included here because it was monitored whether the content changes during preservation and not how it changes after the preparation of jams

  1. Stevia and saccharine are alternative sweetener that have much stronger sweetness than conventional sugar. Would you provide a recipe for each of jam? The real recipe could affect the state of anthocyanins in the jams.

R: The jams were prepared taking into account the sweetening degree of each sweeteners, as is mention now in the manuscript. Indeed, the recipe influences the anthocyanins content, but how much the anthocyanins degrade was expressed as a percentage, precisely so that a comparison between jams can be made that is not affected by what was initially.

  1. Tables, you should provide statistical analysis results.

R: Statistical analysis results was inserted in the supplementary material in order not to load the table too much.

  1. Provide standard curve of acarbose inhibition effects on the a-glucosidase. By the way, what kind of a-glucosidase did you use?

R: Provided in supplementary material. a-glucosidase are from rice and I mentioned in the text.

  1. Clearly indicate jams in what storage conditions were used in each in vitro and vivo experiments.

R: Jams used in experiments were stored under refrigeration (4°C).

  1. In cell viability assay, jam extracts, 1, 7 and 8, did not show dose dependent behavior for Caco-2 cells. How could you explain this? It is difficult to say this as inhibition by the extract.

R: We agree with the reviewer and we insert an explanation                                       

  1. In PK study, I could not believe the std deviations in each group.  Check the data again.

R: I once again checked the standard deviations and they are calculated correctly.

  1. Methods descriptions are unacceptable quality.

R: Additional explanations were added in the description of the methods

English - There must be some improvement on the descriptions.

R: Done

Reviewer 3 Report

this is a very well permormed study. I only suggest adding a limitation section before the final remark in discussion section.

the quality is ok

Author Response

Thanks for your appreciation.

We insert a limitation remarks.

Reviewer 4 Report

The manuscript of T. Scrob et al. investigates the effects of different sweeteners on the in vitro and in vitro biological activities of anthocyanins of the lingonberry jams. The argument, even if not entirely innovative, could be of interest. However, there are some weaknesses.
Major Concerns:
Lines 46-49: The Authors must clarify whether the problem is the stability of the molecules in question or their absorbability. The sentence should be rewritten.
The sentences of lines 55 and 56 are not well connected;
Table 1: Statistical significance should be highlighted, so it would be easier to understand the meaning of the sentence 123-124.
lines 131-135: a citation should be added. In any case, why perform experiments if results are expected and already described in the literature?
The graphical presentation of Figure 2 with unique axes for different chromatograms is misleading.
lines 152-153: the hypothesis that the observed effect is linked only to the activity of the water is weak but, above all, to confirm it, the Authors should quantify the data;
lines 156-163: How do the authors explain the different effects of sweeteners on the stability of specific anthocyanins? Always with the water activity?
Lines 172-173: if the Authors hypothesize an effect of the different sweeteners on the pH, they should have stabilized this parameter.
Figure 4: it is not clear, the caption should be more explanatory
Lines 311-318: can the Authors give some explanations for the biological effects on the cell lines?
Lines 361-363: it is not clear, and it is not even well explained in materials and methods, how the blood sampling was done. As a consequence of hepatic metabolism, the presence of a substance in the gastric vein does not imply that it could arrive in the systemic circulation.
Lines 377-379 and lines 390-395: the Authors should explain how sweeteners can influence the absorption of anthocyanins.
Minor concerns:
The manuscript presents typographical errors (e.g. lines 31, 63, 314).
Lines 643 and 646: the format of the paragraph must be corrected.

Author Response

Thanks for your suggestion.

Major Concerns:
Lines 46-49: The Authors must clarify whether the problem is the stability of the molecules in question or their absorbability. The sentence should be rewritten.

R: The sentence was rewritten.

The sentences of lines 55 and 56 are not well connected;

R: There are two different paragraphs that present, on one hand, aspects regarding the bioavailability of anthocyanins and, on the other hand, anthocyanins capacity to inhibit digestive enzymes like α-glucosidase.

Table 1: Statistical significance should be highlighted, so it would be easier to understand the meaning of the sentence 123-124.

R: Statistical analysis was highlighted by inserting the results in the supplementary material.

lines 131-135: a citation should be added. In any case, why perform experiments if results are expected and already described in the literature?

R: A citation was added.

The graphical presentation of Figure 2 with unique axes for different chromatograms is misleading.

R: I considered that this presentation is more suggestive for a comparative view. In addition, the axes were aligned.

lines 152-153: the hypothesis that the observed effect is linked only to the activity of the water is weak but, above all, to confirm it, the Authors should quantify the data;

R: The hypothesis does not refer to the results of our study, but to the results of the cited research [15].

lines 156-163: How do the authors explain the different effects of sweeteners on the stability of specific anthocyanins? Always with the water activity?

R: Different effects of sweeteners on the stability of specific anthocyanins are not related to water activity, but to food matrix effect.

Lines 172-173: if the Authors hypothesize an effect of the different sweeteners on the pH, they should have stabilized this parameter.

R: Total acidity of all jams was determined in our previous research, cited in the lines. No jam recipe has an additive that stabilizes the pH. Moreover, acidity is one of the parameters that are mentioned in the quality analysis of jams

Figure 4: it is not clear, the caption should be more explanatory

R: The caption of Figure 4 was changed.

Lines 311-318: can the Authors give some explanations for the biological effects on the cell lines?

R: As suggested, new information was added to the text

Lines 361-363: it is not clear, and it is not even well explained in materials and methods, how the blood sampling was done. As a consequence of hepatic metabolism, the presence of a substance in the gastric vein does not imply that it could arrive in the systemic circulation.

R: We reformulated and insert some information on the text.

Lines 377-379 and lines 390-395: the Authors should explain how sweeteners can influence the absorption of anthocyanins.

R: We tried to explain how sweeteners can influence the absorption of anthocyanins

Minor concerns:
The manuscript presents typographical errors (e.g. lines 31, 63, 314).
Lines 643 and 646: the format of the paragraph must be corrected.

R: Done

Round 2

Reviewer 2 Report

The authors tried solving the concerns raised by reviewers. However, there are still some major problems.

1.  Samples used for the PK or in vitro assay, did you use just after production of jams? or the samples were kept for some time at 4oC?

2. Initial anthocyanin contents are important here. The reliability of in vivo, vitro results are dependent on the initial contents of anthocyanin. Your results indicate that at least 20 ug/mL of anthocyanins were detected in blood.

3. Could you explain why your PK results showed clearly small deviations from other results?   What were the LOD and LOQ of each anthocyanins? 

There are many typos. Check them all.

Author Response

  1. Samples used for the PK or in vitro assay, did you use just after production of jams? or the samples were kept for some time at 4oC?

R: Both the jams and their extracts were freshly prepared before each test as it is mentioned in the text.

  1. Initial anthocyanin contents are important here. The reliability of in vivo, vitro results are dependent on the initial contents of anthocyanin. Your results indicate that at least 20 ug/mL of anthocyanins were detected in blood.

R: The concentration of each anthocyanin in the extracts of jams administered to rats are presented in suplementary material. Anyway, these concentrations were used for the calculation of bioavailabilities in percentages. The concentration of determined anthocyanins were µM.

  1. Could you explain why your PK results showed clearly small deviations from other results?   What were the LOD and LOQ of each anthocyanins?

R: These were the experimentally obtained values. Small deviations are obtained both in studies on homogeneous cell cultures and on homogeneous batches of animals. The cells/animals are similar and the determinations have very close values. Because the HPLC method was not developed by us but was taken from another article that determines anthocyanins in lingonberry juice, as mentioned in the text, LOD and LOQ were not calculated. Anthocyanins were determined with the internal standard method using cyanidin as internal standard compound, not based on calibration curves.

Reviewer 4 Report

The Authors responded exhaustively to all the comments of the Referees. In its present form, the manuscript is acceptable for publication.

Author Response

Thank you for appreciation.